# Air Pollution Characteristics during the 2022 Beijing Winter Olympics

**DOI:** 10.3390/ijerph191811616

**Published:** 2022-09-15

**Authors:** Fangjie Chu, Chengao Gong, Shuang Sun, Lingjun Li, Xingchuan Yang, Wenji Zhao

**Affiliations:** 1School of Resources, Environment & Tourism, Capital Normal University, Beijing 100048, China; 2School of Civil and Architectural Engineering, Shandong University of Technology, Zibo 255000, China; 3Beijing Municipal Ecological and Environmental Monitoring Center, Beijing 100048, China

**Keywords:** 2022 Winter Olympics, air pollutants, air quality, particulate matter, influencing factor, combined indicators, pollution control effectiveness

## Abstract

Using air pollution monitoring data from 31 January to 31 March 2022, we evaluated air quality trends in Beijing and Zhangjiakou before and after the 2022 Winter Olympics and compared them with the conditions during the same period in 2021. The objective was to define the air quality during the 2022 Winter Olympics. The results indicated that: (1) the average concentrations of PM_2.5_, PM_10_, NO_2_, CO, and SO_2_ in Zhangjiakou during the 2022 Winter Olympics were 28.15, 29.16, 34.96, 9.06, and 16.41%, respectively, lower than those before the 2022 Winter Olympics; (2) the five pollutant concentrations in Beijing showed the following pattern: during the 2022 Winter Olympics (DWO) < before the 2022 Winter Olympics < after 2022 Winter Paralympics < during 2022 Winter Paralympics; (3) on the opening day (4 February), the concentrations of the five pollutants in both cities were low. PM_2.5_ and PM_10_ concentrations varied widely without substantial peaks and the daily average maximum values were 15.17 and 8.67 µg/m^3^, respectively, which were 65.56 and 69.79% lower than those of DWO, respectively; (4) the PM_2.5_ clean days in Beijing and Zhangjiakou DWO accounted for 94.12 and 76.47% of the total days, respectively, which were 11.76 and 41.18% higher than those during the same period in 2021; (5) during each phase of the 2022 Winter Olympics in Beijing and Zhangjiakou, the NO_2_/SO_2_ and PM_2.5_/SO_2_ trends exhibited a decrease followed by an increase. The PM_2.5_/PM_10_ ratios in Beijing and Zhangjiakou were 0.65 and 0.67, respectively, indicating that fine particulate matter was the main contributor to air pollution DWO.

## 1. Introduction

In recent years, China has hosted several major international events, such as the Beijing Olympic Games, Guangzhou Asian Games, and Asia-Pacific Economic Cooperation (APEC) conference; therefore, it has carried out a series of short-term control measures to ensure air quality, with obvious results. Wu et al. [1] analyzed changes in atmospheric particulate matter concentrations in Beijing during and after the Olympics and their main influencing factors, and they found that the PM_2.5_ and PM_10_ concentrations in Beijing during the Olympics were 18.2 and 16.0% lower than those after the Olympics, respectively, and that local source emissions and regional transport substantially affected particulate matter concentrations. Wang et al. [2] found that ambient concentrations of traffic-related NOx and VOCs at urban sites dropped by 25% and 20–45% in the first two weeks after full control was put. The favorable meteorological conditions during the Beijing Olympics also had a positive impact on primary and secondary pollutant concentrations. Significant decreases in major air pollutant concentrations indicate that the pollution control measures adopted during the 2008 Olympic Games were effective in improving air quality, and the strong variations of PM_2.5_ over the three years imply that special measures taken for traffic control can be considered as a very effective measure of decreasing PM_2.5_ in suburban areas [3]. Studies on the pollution scenarios in Beijing and the surrounding area during the APEC conference found that different levels of control measures (e.g., suspending production and restricting heavy polluting industrial enterprises, restricting motor vehicles, and strictly controlling construction dust) had larger impacts on pollutant changes compared to meteorological conditions [4,5,6,7] used statistical analyses to evaluate the effects of emission reduction during the APEC meeting and reported that the average concentrations of PM_2.5_, PM_10_, SO_2_, and NO_2_ were decreased by 45%, 43%, 64%, and 31% compared to those in the same period of the last 5 years, and a significant reduction in peak PM_2.5_ concentrations. However, the meteorological conditions and pollutant emissions during the 2022 Beijing Winter Olympics were different from these earlier events.

The 24th Olympic Winter Games and the 13th Paralympic Winter Games were held in Beijing and Zhangjiakou from 4–20 February and from 4–13 March 2022, respectively. The main venues were the Beijing Olympic Center, Beijing Shougang Park, Beijing Yanqing, and Zhangjiakou Chongli. The Beijing–Tianjin–Hebei region is an important political, cultural, and economic area of China that is located in the western part of the Bohai Sea region with a dense population and high level of urbanization. The contribution of local pollution to the total is 56–72%, which is the main cause of pollution in the Beijing–Tianjin–Hebei region [8,9]. Since the 2008 Beijing Olympics, a series of treatment measures conducted in and around Beijing have promoted continuous improvements in air quality. The Beijing administration has created novel air pollution treatments for megacities that have resulted in improved air quality in the region. Seasonal variations in pollutant concentrations in the Beijing–Tianjin–Hebei region show that the heaviest pollution occurs during the winter, followed by spring, autumn, and summer [10,11,12]. Due to the unfavorable pollution dispersion conditions, it was challenging to guarantee the air quality during the 2022 Winter Olympics and Paralympics. Researchers analyzed air pollution characteristics during the same period in the year before that of the 2022 Beijing Winter Olympic and Paralympic Games. Li et al. [13] found that heavy pollution occurred for 2–9 d in Beijing during the same period as that of the 2022 Winter Olympics and that the average winter PM_2.5_ concentration in Zhangjiakou during the last 5 years was 30.8 µg/m^3^, with a low anthropogenic air pollutant emission intensity. From these findings, they predicted that the 2022 Winter Olympics would have a background of clean winter air environment. Pan [14] found that the pollution frequency and extent during the same period as that of the 2022 Winter Olympics was higher in Beijing than in Zhangjiakou and that pollutant emissions from Beijing, Tianjin, Hebei, and the surrounding cities were reduced by 50–75%; thus, heavy pollution days might not occur during this period in 2022. Chen et al. [15] studied air pollution data from 2014 to 2019 and found that the air quality improved overall in both cities and that Zhangjiakou’s air quality was better than that of Beijing, and its emissions compliance rate of PM_2.5_ was over 80%. SO_2_ concentrations in Zhangjiakou were the most significantly reduced; however, the PM_2.5_ and PM_10_ concentrations increased. Thus, managing particulate matter pollution in Zhangjiakou was an important management strategy.

Currently, most studies have focused on predicting the air quality during the 2022 Winter Olympics, whereas fewer studies have investigated the actual changes in atmospheric pollutant concentrations during the 2022 Winter Olympics. Therefore, using air pollutant data from 12 national control sites in Beijing and 5 in Zhangjiakou, we compared and analyzed the changes in air pollutant concentrations in the two host cities during the periods before, during, and after the Olympic Games and determined the effects of meteorological conditions and pollution prevention and control measures on air quality. We also evaluated the air quality improvement after emission reduction measures, to provide a reference for future national joint prevention and control measures. The 2022 Winter Olympic Games attracted attention to air pollution in Beijing and Zhangjiakou. This paper aims to provide a scientific assessment of how air quality changed during the 2022 Winter Olympic Games and to help explore long-term mechanisms for air quality improvement in China.

## 2. Data and Methods

### 2.1. Data Acquisition

The study period was 1 January–31 March of 2019–2022. The data for five air pollutant monitoring parameters (PM_2.5_, PM_10_, SO_2_, NO_2_, and CO) used in this study were obtained from state-controlled stations released by the China General Environmental Monitoring Station (http://www.cnemc.cn/) (accessed on 1 May 2022). Twelve monitoring stations in Beijing (Wanshou Xigong, Dingling, Dongsi, Temple of Heaven, Agricultural Exhibition Center, Guangyuan, Haidian Wanliu, Shunyi New Town, Huairou Town, Changping Town, Olympic Sports Center, and Old Town) were used to represent the pollution level in Beijing. Average pollutant concentrations from five monitoring stations in Zhangjiakou (People’s Park, Tanji Factory, Wukinbank, Century Haoyuan, and North Pump House) were used to represent the pollution level in Zhangjiakou. The specific locations are shown in Figure 1. The mass concentration of PM_2.5_ was measured using a Thermo Fisher 1405F monitor with a tapered element oscillating microbalance method (TEOM). The PM_10_ monitor was a Thermo Fisher 1400 monitor (Thermo Fisher, Waltham, MA, USA), also based on the TEOM method. NO_2_ was analyzed using a Thermo Fisher 42C (Thermo Fisher, Waltham, MA, USA) chemiluminescent NO-NO_2_-NOx analyzer with a minimum detection limit of 0.05 × 10^−9^ (volumetric fraction). SO_2_ was analyzed using the Thermo Fisher 43i pulsed UV fluorescence method. Data checking and outlier handling were performed. Zero or negative values that appeared in the case of instrument failure, unstable operation, or uncontrolled environmental quality were considered invalid and were not included in the statistical analysis.

Hourly surface meteorological data, including wind speed (WS), wind direction, temperature, and relative humidity (RH), were obtained for Beijing and Zhangjiakou from 1 January to 31 March of 2021 and 2022 from the National Climate Data Center website (https://www.ncei.noaa.gov/ (accessed on 10 May 2022)). Ground-based observations were contemporaneous with air pollution data.

### 2.2. Methodology

#### 2.2.1. Time Period Divisions

To assess the impacts of the 2022 Winter Olympics emission reduction measures on air quality, we divided the observation period of 1 January–31 March into 5 periods: before the 2022 Winter Olympics (BWO; 1 January–3 February); during the 2022 Winter Olympics (DWO; 4–20 February); during the interval (21 February–3 March); during the 2022 Winter Paralympics (DWP; 4–13 March); and after the 2022 Winter Paralympics (AWP; 14–31 March). By analyzing the characteristics of pollutant concentrations at different time periods, the changes in pollutant concentrations during the event were elucidated.

#### 2.2.2. Correlation Analysis

Pearson correlation coefficients between air pollutants and meteorological factors were calculated (Equation (1)) to determine the relationships between meteorological conditions and air pollutant concentrations.
(1)$$r=\frac{\sum_{i=1}^{n}\left(X_{i}-\bar{X}\right)\left(Y_{i}-\bar{Y}\right)}{\sqrt{\sum_{i=1}^{n}{\left(X_{i}-\bar{X}\right)}^{2}}\sqrt{\sum_{i=1}^{n}{\left(Y_{i}-\bar{Y}\right)}^{2}}}$$
where *r* is the Pearson correlation coefficient; *n* is the number of arrays used in the correlation analysis; and *x* and *y* are eigen values, where *i* = 1, 2,..., and *n* is the number of objects.

## 3. Results and Discussion

### 3.1. Overall Changes in Pollutant Concentrations

To study the air quality characteristics before, during, and after the 2022 Winter Olympics in the host cities (Beijing and Zhangjiakou) and to explore the influences of meteorological conditions and pollution control measures on air quality, we compared the pollutant concentrations during this period with the daily average air pollutant concentrations during the same period in 2019–2021 (Figure 2).

Air quality has improved in recent years, with significant reductions in the concentrations of all five pollutants (PM_2.5_, PM_10_, SO_2_, NO_2,_ and CO) DWO. Compared with the same period in 2019, 2020, and 2021, the pollutant concentrations in 2022 were 17.13–58.34%, 33.52–70.04%, and 38.08–65.80% lower, respectively. PM_2.5_ and SO_2_ concentrations decreased considerably (31.27–70.04% and 40.21–58.34%, respectively), while that of NO_2_ had the smallest decrease (17.13–38.08%). The NO_2_ concentrations observed DWO were substantially lower compared to the same period in 2021 and the effect of control of emissions from mobile sources during the games was outstanding. Strengthened air pollution controls in Beijing, Tianjin, Hebei, and neighboring cities during 2019–2021, including energy structure adjustment, bulk coal control, and ultra-low emission measures in the iron and steel industries, led to substantial changes in air pollutant emissions in the Beijing–Tianjin–Hebei region, including considerable reductions in PM_2.5_ and SO_2_ emissions [16,17].

The pollutant concentrations in both Beijing and Zhangjiakou DWO decreased compared to those BWO. The average NO_2_ concentration in Beijing BWO (1 January–3 February) was 33.53 µg/m^3^, which was similar to the average value for the same periods in 2019–2021 (38.53 µg/m^3^). The average NO_2_, CO, PM_10_, PM_2.5_, and SO_2_ concentrations DWO (4–20 February) were 18.89 µg/m^3^, 0.42 mg/m^3^, 35.08 µg/m^3^, 22.82 µg/m^3^, and 2.41 µg/m^3^, respectively, representing decreases of 43.67, 38.79, 27.18, 45.51, and 13.52%, respectively, compared with those BWO. PM_2.5_ and NO_2_ concentrations decreased the most, while the SO_2_ concentration decreased the least compared to those before the 2022 Winter Olympics. The SO_2_ concentrations in Beijing are low and mainly influenced by regional transport (65% contribution) [18]; thus, the observed decrease was not significant. PM_2.5_ and NO_2_ concentrations decreased mainly due to controls on traffic and industrial sources to reduce pollution emissions during the Olympics. Favorable meteorological conditions also played a positive role in diffusing pollutants. Previous studies have shown that the number of heavy trucks is significantly positively correlated with particulate matter and NO_2_ concentrations in Beijing and that the demand for inter-provincial cargo turnover is an important factor in the increasing number of heavy trucks [19]. Therefore, truck restrictions were adopted during the games, causing the cargo turnover in Beijing to decrease by 31.10% in February 2022 compared to January and by 11.70% compared to the same period in 2021, according to data from the Beijing Municipal Bureau of Statistics. In addition, compared to the periods BWO and DWP, the temperature and RH in Beijing during the Olympics decreased by 10.4–20.4%, and the mainly northerly average wind speed increased by ~21.7%. The favorable meteorological conditions dispersed pollutants and diluted those with lower concentrations, which ensured good air quality DWO [20].

During the study period, PM_2.5_ and PM_10_ concentrations in Beijing increased in the following order: DWO < BWO < interval < AWP < DPW. The concentrations of the other pollutants increased in the order of DWO < BPW < AWP < interval < DWP. All five pollutants (PM_2.5_, PM_10_, NO_2_, CO, and SO_2_) reached their highest concentrations DPW (60.36 µg/m^3^, 107.58 µg/m^3^, 36.30 µg/m^3^, 0.71 mg/m^3^, and 4.06 µg/m^3^, respectively), which were 164.43, 206.66, 92.23, 69.33, and 68.60% higher than those DWO, with PM_10_ and PM_2.5_ exhibiting the largest increases. Owing to southerly winds in March, unfavorable diffusion conditions developed and northerly winds transported sand and dust, among other effects. Compared with the same period in previous years, the pollutant concentrations DWO were considerably lower. The average wind speed in Beijing DWO reached 2.84 m/s, which was 5.54–14.66% faster than during the same period in previous years. The average temperature was −2.63 °C, which was 0.8–2.4 °C lower than those in previous years. Thus, the meteorological conditions in 2022 were slightly more favorable for diffusing pollutants compared with conditions in previous years.

The average concentrations of PM_2.5_, PM_10_, NO_2_, CO, and SO_2_ in Zhangjiakou DWO were 22.34 µg/m^3^, 33.22 µg/m^3^, 12.83 µg/m^3^, 0.57 mg/m^3^, and 6.66 µg/m^3^, which were 28.15, 29.16, 34.96, 9.06, and 16.41%, respectively, lower than those BWO. NO_2_ exhibited the largest decrease. Official information released by the 2022 Beijing Winter Olympics Organizing Committee indicates that the percentage of energy-saving and clean energy vehicles used for transportation services for the 2022 Winter Olympic Games was the highest ever. Along with controls on mobile sources, NO_2_ concentrations decreased as a result. Throughout the observation period in 2022, the NO_2_ and PM_10_ concentrations in Zhangjiakou City were the lowest during the Olympics. The low and variable CO concentrations observed during all phases of the observation period were mainly due to interactions between temperature changes (heating), control measures, and pollution processes. The SO_2_ concentrations during all phases decreased the most compared to the same period in 2019–2021. Industrial combustion, residential combustion, and industrial processes are the main sources of SO_2_ emissions in the city [21,22], and controls targeting these aspects DWO also led to substantial reductions in its emissions.

Comparing the two cities, the decreases in CO and SO_2_ concentrations in Zhangjiakou city were ~17.67% higher than those in Beijing during the last four years, while the decrease in PM_2.5_ concentrations in Beijing was ~20.10% higher than in Zhangjiakou. Zhangjiakou is located in the northwest part of Hebei Province, which is the intersection of the Beijing–Tianjin–Hebei province and the Mongolia economic circle. Figure 2 shows that all pollutants except SO_2_ had lower overall concentrations in Zhangjiakou than in Beijing during 2019–2022. Zhangjiakou City has a smaller population, a smaller industrial scale, fewer contributions from industrial emission sources, and a larger share of primary industries [23]; thus, the impact of industrial emission reductions on CO and SO_2_ pollution in Zhangjiakou City was more substantial in recent years.

### 3.2. Daily Changes in Pollutant Concentrations

To study the impacts of emission reduction measures on the air quality in the two cities, we compared the pollutant concentration levels before and after the Olympics with the daily average air pollutant concentrations during the same period in 2021 (1 January–31 March 2021), the results of which are shown in Figure 3. The PM_2.5_ and PM_10_ concentrations in Beijing were 41.89 and 48.17 µg/m^3^, respectively, BWO. The PM_2.5_ concentration in 2022 was slightly higher (8.64%) and the PM_10_ concentration was slightly lower (−32.67%) than those during the same period in 2021. The PM_2.5_ and PM_10_ concentrations in Zhangjiakou were 31.09 and 46.90 µg/m^3^, respectively, both of which were lower than those during the same period in 2021 (−20.68% and −56.50%, respectively). According to the ***Ambient Air Quality Standard (GB3095-2012)***, the number of clean PM_2.5_ and PM_10_ days in Beijing BWO accounted for 52.94% of the total number of days. The number of clean PM_2.5_ days accounted for 5.88% less of the total number of days during the same period in 2021. Although PM_10_ increased by 17.65% during the same period in 2021, only 4 and 0 heavy pollution days and 18 and 16 pollution days occurred, respectively. The number of PM_2.5_ and PM_10_ clean days in Zhangjiakou accounted for 76.47 and 70.59% of the total number of days, which was 8.82 and 29.41% higher than during the same period in 2021, while the numbers of heavy pollution days accounted for 0 and 8.83%, respectively. The percentage of clean days in Zhangjiakou was higher than in Beijing and the number of polluted days was ~40% lower than in Beijing, indicating that the overall particulate matter concentrations in Zhangjiakou were lower than in Beijing BWO. The NO_2_ concentrations in both cities were relatively unchanged compared to the same period in 2021, and NO_2_ concentrations decreased after January 25 due to the Spring Festival holiday.

DWO (4 February–13 March 2022), some pollutant concentrations exhibited an initially decreasing and then increasing trend, with the lowest concentrations occurring DWO (4–20 February). On the opening day of the Olympics (February 4), the PM_2.5_, PM_10_, CO, NO_2_, and SO_2_ concentrations reached very low values in Beijing (13.82 µg/m^3^, 20.13 µg/m^3^, 7.40 µg/m^3^, 0.47 mg/m^3^, and 6.09 µg/m^3^, respectively). The PM_2.5_ and PM_10_ concentrations in Beijing changed slowly from 4 February to 7 February, then increased slightly after the Chinese New Year holiday on 8 February, peaking on 10 February (55.17 and 78.77 µg/m^3^, respectively). After 24 February, the PM_2.5_ and PM_10_ concentrations increased, peaking on March 10 (199.90 and 253.54 µg/m^3^, respectively). The PM_2.5_ and PM_10_ concentrations in Zhangjiakou were consistent with those in Beijing and peaked on 10 February (55.99 and 77.53 µg/m^3^, respectively), but increased less than those in Beijing after 24 February and peaked on 10 March (67.56 and 143.26 µg/m^3^, respectively), which were 57.82 and 43.50% lower than those in Beijing, respectively. The amount of PM_2.5_ and PM_10_ clean days accounted for 68.42 and 52.63% of the total, respectively, DWO in Beijing, and only three heavy pollution days occurred. PM_2.5_ clean days accounted for 76.47% of total days DWO, which was 41.18% higher than during the same period in 2021, and no heavy pollution days occurred. The PM_2.5_ and PM_10_ clean days in Zhangjiakou DWO accounted for 89.47 and 71.05% of the total days, respectively, and zero and one heavy pollution days occurred, respectively. The PM_2.5_ clean days accounted for 94.12% of the total days DWO, which was 11.76% higher than those during the same period in 2021, and no heavy pollution days occurred. During the Olympics, regional joint prevention and controls were effective, with average PM_2.5_ concentrations in Beijing and Zhangjiakou of 35.55 and 24.17 µg/m^3^, respectively, which were 57.61 and 38.30% lower than those in 2021. In addition, the number of heavy pollution days decreased by 83 and 100% compared to 2021. Thus, the number of clean days in Zhangjiakou DWO and DWP was approximately 20% more than in Beijing during the same period. Only one heavy pollution day occurred and the concentrations of particulate matter were lower than those in Beijing, resulting in better air quality in Zhangjiakou.

The NO_2_ concentration in Beijing DWO decreased by 38.08% compared to 2021, indicating a substantial reduction in emissions. The maximum NO_2_ concentrations in Beijing and Zhangjiakou DWO occurred during 9–10 March (56.01 and 30.50 µg/m^3^, respectively), which were 41% and 19.79% higher than the maximum concentrations observed DWO. The NO_2_ increase in Zhangjiakou City was much lower than in Beijing according to data from the Beijing Municipal Bureau of Statistics and the Hebei Provincial Bureau of Statistics. At the end of 2021, Beijing’s motor vehicle fleet was approximately five times larger than that of Zhangjiakou. Therefore, Beijing’s emissions from mobile sources contributed more to pollution. The average NO_2_ concentration in Zhangjiakou City from January to March 2022 was 30.01% lower than the annual average for 2021 and 34.16% lower than the same period in 2021. NO_2_ concentrations in both cities remained relatively low DWO.

The CO concentrations in Beijing DWO decreased by 52.95% compared to 2021 levels, indicating that CO emission reductions in Beijing were more effective. The CO concentrations in Zhangjiakou decreased by 21.73% compared to 2021 levels, which was consistent with the changes in NO_2_. CO emissions in urban areas are derived from factory production and vehicle exhaust emissions, both of which are closely related to human activity [24]. Unfavorable diffusive conditions during winter increase the likelihood of CO accumulation [25]; thus, the CO reductions observed here were likely influenced by industrial and transportation emission reduction measures.

The average SO_2_ concentrations in Beijing and Zhangjiakou DWO and DWP were 7.39 and 2.86 µg/m^3^, respectively, which do not constitute serious pollution levels. The average daily concentrations were also below the national ambient air quality class I standard (50 µg/m^3^). The average SO_2_ concentration in Beijing from 1 January to 31 March in 2022 was 20.59% lower than in 2021, with the largest reduction (46.33%) occurring DWO. The SO_2_ concentration in Zhangjiakou DWO was 60.55% lower than in 2021 and did not rebound after the end of the 2022 Winter Olympics. SO_2_ concentrations are influenced by population density and the proportion of secondary industries, in addition to temperature and vegetation index [26]. Zhangjiakou has higher SO_2_ emissions during the heating season, whereas its low winter temperatures and precipitation have less of an effect on SO_2_ removal [27]; thus, SO_2_ reductions were mainly influenced by source emission controls DWO. Comparing the concentration curves of both cities, the overall SO_2_ concentrations in Zhangjiakou were higher than those in Beijing, owing to the influence of winter coal heating and industrial emissions. The overall SO_2_ concentrations in Zhangjiakou were 333.08% and 146.86% higher than those in Beijing during the entire observation period in 2021 and 2022, respectively.

The pollutant concentrations decreased after emission controls were implemented in both regions DWO. To study the effect of pollutant reduction DWO, we compared the average pollutant concentrations in Beijing and Zhangjiakou DWO, DWP, and during the Spring Festival with concentrations during the same periods in 2021. The pollutant reduction percentages are listed in Table 1. Firework displays and increased coal consumption during the Spring Festival produce significant increases in PM_2.5_, PM_10_, and SO_2_ concentrations [28,29,30,31,32]. During the 2022 Spring Festival, Beijing and the eight surrounding provinces implemented a strict ban on the sale and use of fireworks. As a result, the PM_2.5_ concentrations in Beijing and Zhangjiakou decreased by 76.69 and 48.75% compared to 2021, while SO_2_ concentrations decreased by 38.38 and 57.81%, respectively, both of which were the best levels observed since monitoring began. During the Olympics, temporary controls were adopted for some industries and vehicles that have high pollutant emissions with relatively low economic impacts. The 2022 Winter Olympics also coincided with the Spring Festival. As some businesses closed for the holidays, the level of social and economic activities in the region decreased considerably and the traffic flow decreased. A comprehensive assessment found that pollutant emissions in this region decreased by 38.08–65.80%, further contributing to air quality improvement.

To ensure the normal operation of traffic during the Beijing 2022 Winter Olympic Games and Winter Paralympic Games, the municipal government decided to take temporary traffic control. From January to March 2022, Beijing and Zhangjiakou set up traffic lanes reserved for the Winter Olympics with a total length of 239.5 km. During the Winter Olympic Games and the Winter Paralympic Games, in addition to trucks carrying essential goods, other trucks from other provinces needed to detour around the roads in Beijing and Zhangjiakou. From 21 January to 16 March 2022, from 6:00 p.m. to 24:00 p.m. daily, many provincial highways were closed to trucks of 4 tons (not included) or more. There was advocacy for the city’s units to adopt flexible work systems such as home working, telecommuting, and staggered commuting, while guiding green travel (http://jl.people.com.cn/n2/2022/0115/c349771-35096436.html) (accessed on 1 June 2020). That also contributed to air quality improvement.

### 3.3. Daily Pollutant Variations

To further explore the impacts of implementing emission reduction measures on air pollutant concentrations, we calculated the average PM_2.5_, PM_10_, NO_2_, SO_2_, and CO concentrations at different points during the period of the 2022 Winter Olympics, as well as the hourly averages during the same period and 1 January–3 February of 2021 and after the 2022 Winter Olympics (4 February–31 March) (i.e., daily variations in pollutant concentrations). PM_2.5_ and PM_10_ data from two dusty days (15 March and 28 March 28 2021) were excluded. In addition to the five study phases mentioned above, the hourly average pollutant concentrations on the opening day of the 2022 Winter Olympics (4 February 2022) were compared with each phase to determine the impacts of emission reduction measures on air quality on the opening day (Figure 4).

Figure 4 shows that the hourly average pollutant concentrations in Beijing and Zhangjiakou during all phases in 2022 were lower than those during the same period in 2021, with much lower peaks. The daily variations in each pollutant in Beijing from January to March 2021 were large, with considerable bimodal characteristics. The first maximum PM_2.5_ and PM_10_ concentrations occurred at 20:00 local time (92.83 and 157.47 µg/m^3^, respectively). The second maximum occurred at approximately 10:00, when the temperature and humidity were favorable for secondary organic aerosol formation [33,34]. The accumulation process during all phases of the 2022 Winter Olympics became much slower, with only one significant peak in each of the PM_2.5_ and PM_10_ concentrations (44.04 and 28.69 µg/m^3^, respectively) at 22:00, with a peak lag compared to the same period in 2021 and a slower change in the peak magnitude. Both the CO and NO_2_ concentrations during all phases of the same period in 2022 and 2021 exhibited considerable bimodal characteristics. CO and NO_2_ in urban areas are mainly derived from traffic sources, and their concentrations vary with the traffic volume [35,36]. Maxima generally occur from 07:00 to 09:00 and after 21:00 and concentrations reach their minima during the afternoon, which is consistent with the morning and evening maxima. The overall variations in the CO and NO_2_ concentrations DWO were similar to those during the same period in 2021; however, the overall pollutant variation trend slowed down. The minimum value was reached at approximately 07:00, whereas the maximum value was reached at approximately 13:00. Motor vehicle emissions and regional transport are the main sources of SO_2_ in Beijing, and transporting SO_2_ in the inverse thermosphere to the ground under the thermal action of the sun leads to a peak at noon [37]. The maximum SO_2_ concentration DWO was observed one hour later and was 54.79% lower than the concentration recorded during the same period in 2021. In addition, no second peak occurred at night DWO. Nighttime SO_2_ concentrations in Beijing are closely related to the regional transport of SO_2_ emitted by factories in the surrounding area during the daytime [38]. The decreased nighttime SO_2_ concentrations DWO indicate that reduced SO_2_ pollution from factories in the areas surrounding Beijing played a positive role in decreasing SO_2_ pollution.

The daily pollutant variations in Zhangjiakou City were lower than those during the same period in 2021 and the concentration variation was smaller; thus, the peak-shaving effect was more pronounced. The PM_10_, PM_2.5_, CO, SO_2_, and NO_2_ peaks were 53.45, 43.49, 33.41, 50.17, and 64.68% lower, respectively, DWO, and were 38.84, 53.25, 38.37, 71.42, and 46.84% lower, respectively, DWP. The maximum SO_2_ concentrations in Zhangjiakou City were observed at 10:00 and 21:00, and the number of peaks was reduced compared to the same period in 2021. Thus, the peak reduction effect of SO_2_ was the most pronounced. Comparing the daily variation curves for the two cities, the daily average NO_2_ and SO_2_ concentrations varied widely between the cities. The two NO_2_ peaks observed in Zhangjiakou DWO occurred at 08:00 and 19:00 and reached a minimum (8.56 µg/m^3^) at approximately 03:00. Unlike Beijing, the evening peak concentration in Zhangjiakou was 13.67% higher than the morning peak concentration and occurred ~2 h earlier than in Beijing. Beijing restricts trucks from passing between 23:00 and 6:00; thus, NO_2_ emissions are higher at night. The SO_2_ concentrations in Zhangjiakou were higher than those in Beijing, two peaks were observed, and the pattern was consistent during and after the 2022 Winter Olympics. On 4 February 2022 (the opening day of the 2022 Winter Olympics), all five pollutant concentrations in both cities reached their lowest values during the study period. The PM_2.5_ and PM_10_ concentrations varied widely without substantial peaks and the daily average maximum values were 15.17 and 8.67 µg/m^3^, respectively. The maximum values were 65.56 and 69.79% lower than those DWO, respectively. NO_2_ concentrations were lower during the morning peak (77.81 and 84.56% lower than those DWO and the same period in 2021, respectively). The concentration increased after 18:00 and reached a maximum at 23:00. The characteristics of the changes in CO were consistent with those DWO, with more significant double peaks; however, the peak was 61.11% lower than DWO. In summary, the air quality on the opening day was much better than during the other two periods and no obvious pollutant accumulation occurred. Although the meteorological factors on the opening day were favorable for diffusing pollutants [39], the reduced peak concentrations reflect the positive impacts of the control measures. Zhangjiakou City has a relatively small population and has improved its air quality in recent years; therefore, background concentrations are generally low [40,41] and changes in pollutant concentrations on the opening day decreased less than those in Beijing.

### 3.4. Meteorological Influences

Meteorological conditions, including temperature, wind speed, relative humidity, and precipitation, are the main factors that contribute to daily variations in pollutant concentrations [42,43,44]. Figure 5 shows the daily variations in wind speed, air temperature, and relative humidity during the study period in 2022 and the same period in 2021. The overall average temperature during the 2022 observation period in Beijing was 2.21 °C, which was 1.02 °C lower than during the same period in 2021; the average wind speed reached 2.53 m/s, which was 0.07 m/s lower than during the same period in 2021; and the average relative humidity was 44.97%, which was 0.57% lower than during the same period in 2021. The overall average temperature in Zhangjiakou during the study period was −2.28 °C, which was 1.24 °C lower than during the same period in 2021; the average wind speed reached 2.81 m/s, which was 0.16 m/s slower than during the same period in 2021; and the average relative humidity was 43.92%, which was 3.31% lower than during the same period in 2021. Thus, the overall meteorological conditions in 2022 and 2021 were similar.

The meteorological conditions DWO in Beijing and Zhangjiakou were better than those DWP (Table 2). DWO in Beijing, compared to the same period in 2021, the temperature decreased by 158.50%, the RH decreased by 2.17%, and the WS increased by 17.36%. The dominant wind direction on the ground was northerly/northwesterly for much of this time. The overall meteorological conditions were better in 2022 than during the same period in 2021, which positively affected pollutant reduction during the 2022 study period. The temperatures in Beijing DWO increased on average by 5.86 °C, the WS increased by 25.41%, and the RH decreased by 19.67% compared to BWO. The temperatures in Zhangjiakou DWO decreased by 8 °C compared to BWO (−12.80%). The WS and RH increased by 3.07 and 42.41%, respectively, compared to the same period in 2021, and by 23.31 and −2.52% compared to BWO, respectively. The low temperatures, strong cold air activity, and low RH in both host cities were favorable for horizontal pollution diffusion and inhibited pollutant formation through multi-phase reactions [45,46]. Thus, we consider that the meteorological conditions had positive effects on air quality DWO. Weak southwesterly and southeasterly winds in Beijing during winter are important factors that influence heavy pollution [47]. The scenario in the Zhangjiakou region is similar to that in Beijing, where heavy pollution is mainly associated with southwesterly and southerly winds [48]. The frequency of southerly winds in Beijing DWO was ~20%, which was ~10% lower than during the same period in 2021. Two days of southerly winds were observed in Zhangjiakou, which was two days more than during the same period in 2021. Overall, regional transmission transported less pollution, which was conducive to pollutant dilution, removal, and diffusion; thus, the air quality was excellent. In summary, the meteorological conditions DWO fluctuated widely and the pollution diffusion conditions were favorable for good air quality.

We performed a Pearson correlation analysis between three meteorological elements and the five pollutants, the results of which are shown in Table 3. WS was significantly negatively correlated (*p* < 0.01) with NO_2_, CO, PM_2.5_, and PM_10_ during the study period and a decrease in wind speed hindered the diffusion and dilution of atmospheric pollutants both horizontally and vertically. Winter pollution days often occur when the wind speed is less than 3 m/s [49]. The temperature was positively correlated with PM_2.5_ and PM_10_ (*p* < 0.01) and weakly negatively correlated with the other pollutants (*p* < 0.05). The RH was significantly positively correlated with NO_2_, CO, PM_2.5_, and PM_10_ (*p* < 0.01). The WS was significantly negatively correlated with the RH during the study period (*r* = −0.586, *p* < 0.01). A low WS and high RH are conducive to maintaining a steady state in the near-surface atmospheric layer and increasing the inversion intensity, which is unfavorable to the diffusion of pollutants such as PM_2.5_ and PM_10_ both vertically and horizontally and aggravates the accumulation of particulate pollution [50].

The correlation analysis indicates that WS had a strong negative effect on regional air pollutant concentrations. Due to the relatively stable atmospheric structure during winter, a higher WS is more favorable for diffusing and diluting atmospheric pollutants. However, the five pollutants investigated here were not strongly correlated with the average temperature, which is potentially because the study period was too short to reflect the role of temperature on air quality, thereby resulting in a non-significant correlation.

### 3.5. Composite Pollution Characterization

Ratio analysis refers to the use of mass concentration data of the different pollutants to determine the relevant pollution characteristics. It is also used to identify the pollutant sources using their ratios. The ratio analysis is commonly used to characterize atmospheric pollutants using ratios such as NO_2_/SO_2_, PM_2.5_/SO_2_, and PM_2.5_/PM_10_. The pollutant concentration ratios during each phase of the 2022 Winter Olympics and the same period in 2021 were calculated and the results are listed in Table 4.

The PM_2.5_/PM_10_ ratios in Beijing and Zhangjiakou were 21.16 and 30.06% lower than those for the same period in 2021 and for the entire study period in 2022, respectively. The PM_2.5_/PM_10_ ratio in Beijing was 0.65 DWO, which was 25.17% lower than tBWO and 18.69% lower than during the same period in 2021. The PM_2.5_/PM_10_ ratio in Zhangjiakou was 0.67, which was 45.65% lower than BWO and 46.93% higher than during the same period in 2021. These findings indicate that the contributions of coarse particulate matter to air pollution in Beijing increased DWO, and the main pollution source was likely dusty weather [51].

From January to March 2022, the overall PM_2.5_/SO_2_ ratio in Beijing was lower than during the same period in 2021, with an initial decrease followed by an increase. The PM_2.5_/SO_2_ ratio decreased by 36.99% DWO compared to BWO and increased by 56.84% DWP. A high PM_2.5_/SO_2_ value indicates that traffic emissions were the main PM_2.5_ source, and a low value indicates that industrial combustion emissions were dominant [52]. In addition, the average WS decreased, while the temperature and RH increased DWP. The meteorological conditions were more static, producing unfavorable conditions for diffusing pollutants, which also caused the ratios to be higher during the Paralympic Games than during the Olympics. Higher NO_2_/SO_2_ values indicate that the pollution was derived mainly from mobile sources, whereas smaller values indicate that air pollution was derived from stationary sources such as industrial combustion. The changes in the NO_2_/SO_2_ ratio in both cities were consistent with those in the PM_2.5_/SO_2_ ratio, with an initial decrease followed by an increase. The NO_2_/SO_2_ ratio reached the minimum in Beijing DWO, which was 34.86% lower than BWO and 15.39% higher than during the same period in 2021. The NO_2_/SO_2_ ratios DWP in Beijing and Zhangjiakou increased slightly compared to those DWO, decreasing by 41.11 and 1.29%, respectively, compared to the same period in 2021. This indicates that industrial sources decreased considerably and transportation sources increased DWO, indicating a significant effect on reducing pollution from industrial sources in some regions [53]. The changes in the NO_2_/SO_2_ ratio in both cities were consistent with those in the PM_2.5_/SO_2_ ratio. Both exhibited an initial decrease followed by an increase, indicating that traffic emissions in Beijing DWO and the interval period were the lowest throughout the study period. Zhangjiakou city has one-fifth of the motor vehicle ownership of Beijing; thus, basic traffic emissions are lower in Zhangjiakou and the effect of reduction of traffic emissions was less substantial than in Beijing.

In summary, the PM_2.5_/SO_2_ and NO_2_/SO_2_ values DWO for Beijing and Zhangjiakou were lower than those during other phases of the 2022 Winter Olympics, and the smaller ratios indicated a decrease in the contribution of traffic emissions. The PM_2.5_/SO_2_ and NO_2_/SO_2_ values in Beijing were 306.21 and 182.14% higher than those in Zhangjiakou City, respectively. The NO_2_ concentration in Beijing was 18.89 µg/m^3^, which was higher than in Zhangjiakou City, while the SO_2_ concentration was 2.41 µg/m^3^, which was much lower than in Zhangjiakou City, indicating that the emissions from mobile sources in Beijing were higher than those in Zhangjiakou City. Pollutant emissions from industrial sources were low for SO_2_ pollution from stationary sources and were better controlled, resulting in two ratios that were much higher than those of Zhangjiakou City. The PM_2.5_/PM_10_ ratios of both cities DWO were greater than 0.6, although they were lower than those BWO, indicating that although Beijing and Zhangjiakou were more seriously affected by the dusty weather in the north DWO, the contribution of traffic dust pollution was higher compared to sandy dust.

## 4. Conclusions


(1)The concentrations of all five pollutants (PM_2.5_, PM_10_, NO_2_, CO, and SO_2_) in Beijing were in the order of DWO < BWO < AWP < DWP and those in Zhangjiakou city were in the order of DWO < BWO < DWP.(2)DWO, the average concentrations of PM_2.5_ (45.51%) and NO_2_ (43.67%) in Beijing decreased the most compared to the levels BWO, while the SO_2_ concentration (13.52%) decreased the least. NO_2_ exhibited the largest decrease in Zhangjiakou at 34.96%.(3)On the opening day of the Olympics (4 February), the PM_2.5_, PM_10_, CO, NO_2_, and SO_2_ concentrations reached very low values in Beijing (13.82 µg/m^3^, 20.13 µg/m^3^, 7.40 µg/m^3^, 0.47 mg/m^3^, and 6.09 µg/m^3^, respectively). The PM_2.5_ and PM_10_ concentrations varied widely without substantial peaks and the daily average maximum values were 65.56 and 69.79% lower than those DWO, respectively.(4)The frequency of southerly winds in Beijing DWO was ~20%, while only two days with southerly winds were observed in Zhangjiakou. The dominant wind direction on the ground was northerly/northwesterly for much of this time. The overall meteorological conditions were better in 2022 than during the same period in 2021.(5)The PM_2.5_/PM_10_ ratios in Beijing and Zhangjiakou were 0.65 and 0.67, respectively, DWO, which were 18.69 and 46.93% lower than those in the same period in 2021, respectively. This indicates that the contributions of coarse particulate matter to air pollution increased DWO. PM_2.5_/SO_2_ and NO_2_/SO_2_ values for Beijing and Zhangjiakou were lower DWO than those during other phases of the 2022 Winter Olympics, indicating a decrease in the contribution of traffic emissions.


## Figures and Tables

**Figure 1 ijerph-19-11616-f001:**
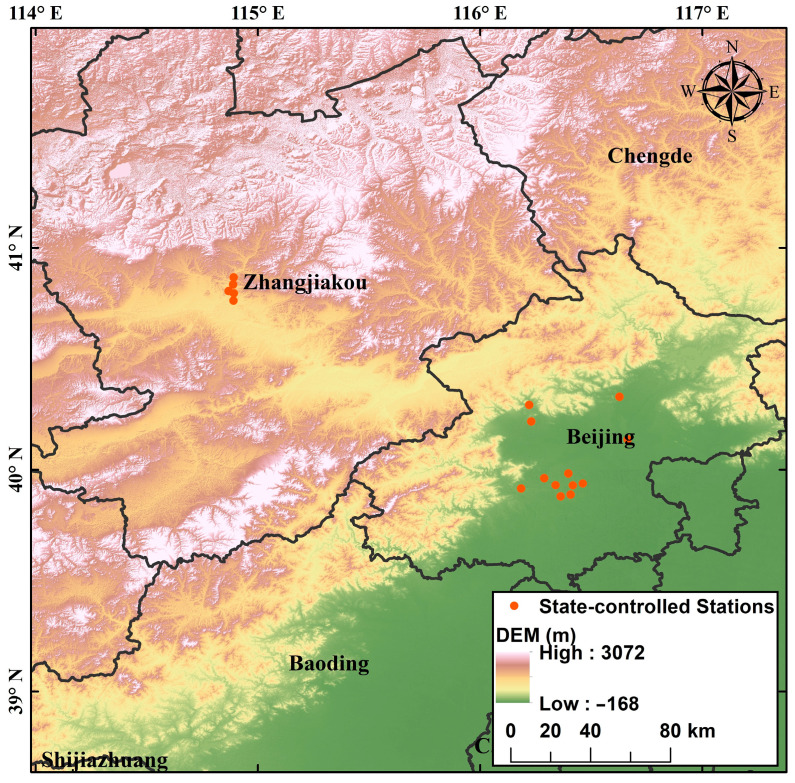
Locations of observation stations in Beijing and Zhangjiakou.

**Figure 2 ijerph-19-11616-f002:**
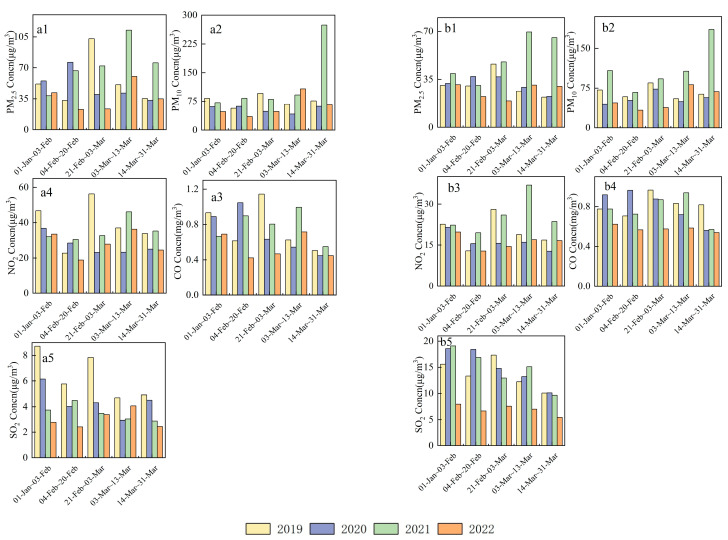
Average pollutant concentrations in (**a**) Beijing and (**b**) Zhangjiakou during different periods. (**a1**–**a5**) show the PM_2.5_, PM_10_, NO_2,_ CO, and SO_2_ concentration in Beijing respectively, (**b1**–**b5**) show the PM_2.5_, PM_10_, NO_2,_ CO, and SO_2_ concentration in Zhangjiakou respectively.

**Figure 3 ijerph-19-11616-f003:**
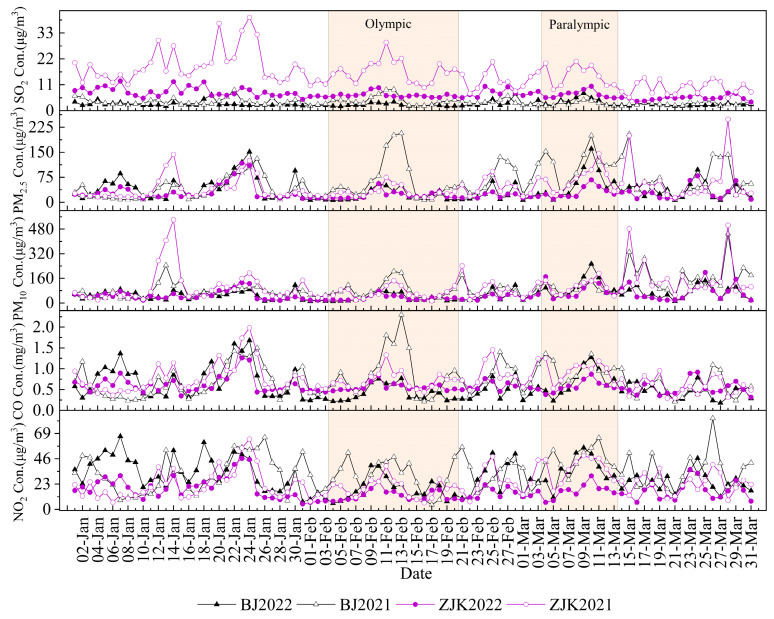
Daily variations in pollutant concentrations in Beijing and Zhangjiakou during the study period.

**Figure 4 ijerph-19-11616-f004:**
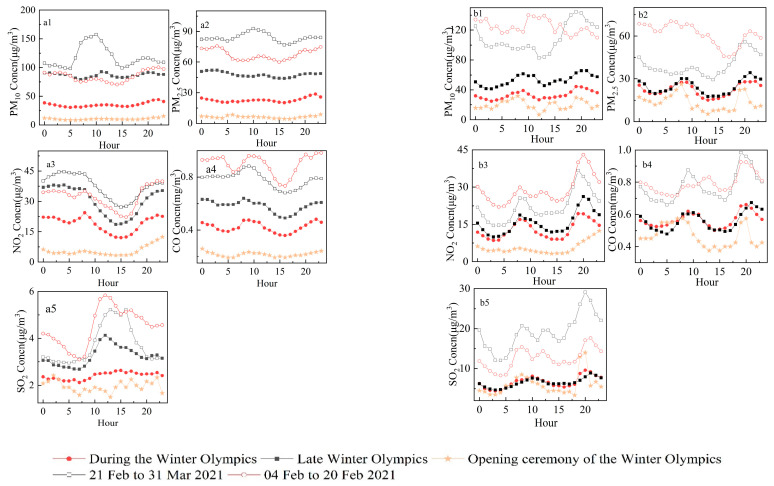
Average pollutant concentrations in (**a**) Beijing and (**b**) Zhangjiakou at different times. (**a1**–**a5**) show the PM_10_, PM_2.5_, NO_2,_ CO, and SO_2_ concentration in Beijing respectively, (**b1**–**b5**) show the PM_10_, PM_2.5_, NO_2,_ CO, and SO_2_ concentration in Zhangjiakou respectively.

**Figure 5 ijerph-19-11616-f005:**
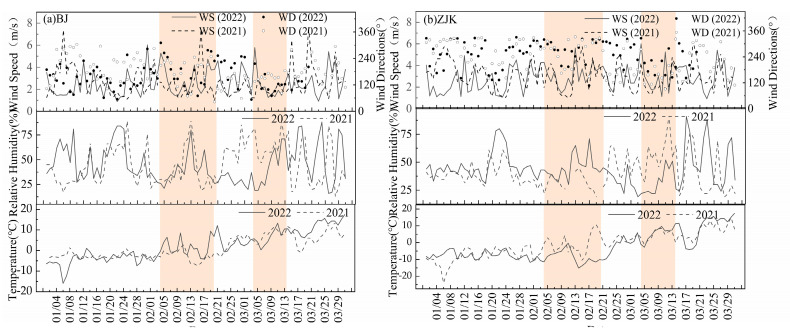
Meteorological conditions DWO compared with those during the same period in 2021. (**a**) Beijing (**b**) Zhangjiakou.

**Table 1 ijerph-19-11616-t001:** Estimated decreases in pollutant concentrations in Beijing and Zhangjiakou.

	Time Period	NO_2_	CO	PM_10_	PM_2.5_	SO_2_
Beijing	Total Average	−16.08%	−23.45%	−52.35%	−42.29%	−20.59%
	DWO	−38.08%	−52.95%	−57.94%	−65.80%	−46.33%
	DWP	−21.16%	−28.15%	17.28%	−46.43%	33.80%
	Spring Festival	−71.82%	−60.19%	−79.19%	−76.69%	−38.38%
Zhangjiakou	Total Average	−30.01%	−22.37%	−54.90%	−41.34%	−54.74%
	DWO	−34.16%	−21.73%	−50.27%	−26.93%	−60.55%
	DWP	−54.14%	−37.61%	−23.66%	−55.68%	−53.54%
	Spring Festival	−56.33%	−21.46%	−73.14%	−48.75%	−57.81%

**Table 2 ijerph-19-11616-t002:** Meteorological data for Beijing and Zhangjiakou during each period.

Period	Beijing			Zhangjiakou		
	Wind Speed (m/s)	Temperature (°C)	Relative Humidity (%)	Wind Speed (m/s)	Temperature (℃)	Relative Humidity (%)
BWO	2.27	−2.61	49.5	2.5	−7.61	46.65
DWO	2.84	−2.63	39.76	3.08	−8.58	45.47
Interval	2.86	3.55	29.45	2.92	−1.66	33.91
DWP	2.3	7.52	43.3	3.01	4.92	32.8
AWP	2.66	7.02	51.72	2.97	9.35	49.61
1 January–31 March 2022	2.53	1.19	44.97	2.81	−2.28	43.92
1 January–31 March 2021	2.6	2.21	45.54	2.97	−1.04	40.61

**Table 3 ijerph-19-11616-t003:** Correlation analysis between the air pollutants and meteorological factors.

Meteorological Factor	NO_2_	CO	PM_10_	PM_2.5_	SO_2_
Wind Speed	−0.485 **	−0.429 **	−0.213 *	−0.362 **	−0.170
Temperature	−0.230 *	−0.220 *	0.279 **	0.024	−0.040
Relative Humidity	0.488 **	0.644 **	0.384 **	0.624 **	0.081

* Significant at the <0.05 level (two-tailed); ** significant at the <0.01 level (two-tailed); *n* = 90.

**Table 4 ijerph-19-11616-t004:** Correlation analysis between air pollutants and meteorological factors in Beijing and Zhangjiakou.

Ratio	Beijing			Zhangjiakou		
	NO_2_/SO_2_	PM_2.5_/PM_10_	PM_2.5_/SO_2_	NO_2_/SO_2_	PM_2.5_/PM_10_	PM_2.5_/SO_2_
BWO	12.04	0.87	15.04	2.48	0.66	3.90
DWO	7.84	0.65	9.48	1.93	0.67	3.36
Interval	8.25	0.48	6.96	1.91	0.50	2.52
DWP	8.94	0.56	14.86	2.41	0.38	4.40
AWP	10.05	0.53	14.42	3.06	0.44	5.51
1 January–31 March 2022	9.99	0.66	12.85	2.39	0.54	3.93
1 January–31 March 2021	9.46	0.54	17.68	1.54	0.42	3.03

## Data Availability

The data presented in this study are available in Table 1, Table 2, Table 3 and Table 4.

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
