# Peer review of "Air Pollution Characteristics during the 2022 Beijing Winter Olympics"

_ijerph, 2022, doi:10.3390/ijerph191811616_

Round 1

Reviewer 1 Report

It is a good paper, with a lot of variables well described. There are many influences and forcings described but the authors manage to achieve their goals.

To improve understanding, it will be good put a separation in figures 2 e 4 where are described Beijing and Zhangjiakou (a and b).

Figures 2 to 5 are displaced, but I suppose it was done when formatting to submit the paper. 

Author Response

Dear Reviewer,

We thank you for the encouraging and insightful comments that have helped us improve our manuscript titled “Air pollution characteristics during the 2022 Beijing Winter Olympics.” We provide below our point-by-point responses to the comments and suggestions of the reviewer. We have carefully addressed all the points raised by you and have made appropriate changes in our manuscript. We hope that our revisions and responses are acceptable and that the manuscript is suitable for publication.

Response 1. Thank you for your valuable suggestion. I have changed the captions of Figures 2 and 4 as follows:

Figure 2. Average pollutant concentrations in (a) Beijing and (b) Zhangjiakou during different periods.

Figure 4. Average pollutant concentrations in (a) Beijing and (b) Zhangjiakou at different times.

Response 2. Thank you for pointing out the displacement in Figures 2 to 5. We have rectified this issue and uploaded the PDF version.

We thank you for your valuable comments that have enabled us to improve our article. We look forward to your positive response.

Regards,

Authors.

Reviewer 2 Report

The paper is well presented with a very accurate data documentation.

The Authors list a very abundant literatur, but only a few titles have been quoted in the text.

I have also noted a relative originality of the paper.

Other comments and proposals are needed.

Author Response

Dear Reviewer,

We thank you for the encouraging and insightful comments that have helped us improve our manuscript titled “Air pollution characteristics during the 2022 Beijing Winter Olympics.” We provide below our point-by-point responses to the comments and suggestions of the reviewer. We have carefully addressed all the points raised by you and have made appropriate changes in our manuscript. We hope that our revisions and responses are acceptable, and that the manuscript is suitable for publication.

Response 1. Thank you for your valuable suggestion. but only a few titles have been quoted in the text. Some research content and perspectives of the literatures have been added in the Introduction section as follows:

Wang et al. (2010) found that ambient concentrations of traffic-related NOx and VOCs at urban site dropped by 25% and 20–45% in the first two weeks after full control was put. The favorable meteorological conditions during the Beijing Olympics also had a positive impact on primary and secondary pollutant concentrations. Significant decreases in major air pollutant concentrations indicate that the pollution control measures adopted during the 2008 Olympic Games were effective in improving air quality, the strongly variations of PM2.5 over the three years imply that special measures taken for traffic control can be considered as a very effective measure of decreasing PM2.5 in suburban areas. (Wang et al., 2014).

Cheng et al. (2016) used statistical analyses to evaluate the effects of emission reduction during the APEC meeting and reported the average concentrations of PM2. 5,PM10,SO2,NO2 were decreased by 45%, 43%, 64% and 31% compared to those in the same period of the last 5 years, and a significant reduction in peak PM2.5 concentrations.

Chen et al. (2021) studied air pollution data from 2014 to 2019 and found that the air quality improved overall in both cities and that Zhangjiakou's air quality was better than that of Beijing, its emissions compliance rate of PM2.5 was over 80%. SO2 concentrations in Zhangjiakou were the most significantly reduced; however, the PM2.5 and PM10 concentrations increased. 

We thank you for your valuable comments that have enabled us to improve our article. We look forward to your positive response.

Regards,

Authors.

Reviewer 3 Report

Thank you for giving me this opportunity to read the manuscript entitled "Air pollution characteristics during the 2022 Beijing Winter Olympics". The topic of this manuscript is interesting and would be a good contribution to this field. I think it could be considered for publication in International Journal of Environmental Research and Public Health once the following issues are addressed.

  1. Please replace the keywords that already appear in the manuscript's title with close synonyms or other keywords, which will also facilitate your paper to be searched by potential readers.

  1. The resolution of Figure 5 should be improved as it is not clear enough to see.

  1. Limitations should be added as a sub-section in the Discussion.

  1. Line3 328-330, Firework displays during the Spring Festival produce significant increases in PM2.5, PM10, and SO2 concentrations (Wang et al., 2016; Xie et al., 2021; Xu et al., 2020). ”: Some newly published paper cloud be cited as references to support the statement here, for example, the paper titled “Dynamic assessment of PM2. 5 exposure and health risk using remote sensing and geo-spatial big data”.

  1. Some grammatical errors exist in the manuscript. Therefore, a critical review of the manuscript language will improve readability.

Author Response

Dear Reviewer,

We thank you for the encouraging and insightful comments that have helped us improve our manuscript titled “Air pollution characteristics during the 2022 Beijing Winter Olympics.” We provide below our point-by-point responses to the comments and suggestions of the reviewer. We have carefully addressed all the points raised by you and have made appropriate changes in our manuscript. We hope that our revisions and responses are acceptable, and that the manuscript is suitable for publication.

Response 1. Thank you for your valuable suggestion. Accordingly, we have added some more keywords.

Response 2. Thank you for pointing this out. We have now improved the resolution of Figure 5.

Response 3:  Thank you for your valuable suggestion. We have added the limitations in Lines 342 and 354 as follows:

To ensure the normal operation of traffic during the Beijing 2022 Winter Olympic Games and Winter Paralympic Games, the municipal government has decided to take temporary traffic control. From January to March 2022, Beijing and Zhangjiakou set up traffic lanes reserved for Winter Olympics while Olympic lanes with a total length of 239.5 kilometers. During the Winter Olympic Games and the Winter Paralympic Games, in addition to trucks carrying essential goods, other trucks from other provinces need to detour around the roads in Beijing and Zhangjiakou. From January 21 to March 16, 2022, from 6:00 p.m. to 24:00 p.m. daily, many provincial highways are closed to trucks of 4 tons (not included) or more. Advocate the city's units to adopt flexible work systems such as home working, telecommuting and staggered commuting, while guiding green travel.  That also contributing to air quality improvement.

Response 4. Thank you for your valuable suggestion. Based on your input, we have modified content and included the citation of Song et al. as follows:

“Firework displays and increased coal consumption during the Spring Festival produce significant increases in PM2.5, PM10, and SO2 concentrations (Song et al., 2019; Ding et al., 2019; Ji et al., 2018; Xie et al., 2021; Xu et al., 2020).” in Lines 328 to 330.

We have added the suggested reference in Lines 645 and 646 as follows:

Song, Y.M., Huang, B., He, Q.Q., Chen, B., Wei, J., Mahmood, R., 2019. Dynamic assessment of PM2.5 exposure and health risk using remote sensing and geo-spatial big data, Environmental Pollution. 253, 288–296.

Response 5. Thank you for your valuable suggestion. Accordingly, we engaged a professional English editing services company and thoroughly checked the manuscript. All language errors have been rectified and the readability has been enhanced.

We thank you for your valuable comments that have enabled us to improve our article. We look forward to your positive response.

Regards,

Authors.